# Prevalence of Older Adult Disability and Primary Health Care Responsiveness in Low-Income Communities

**DOI:** 10.3390/life10080133

**Published:** 2020-08-05

**Authors:** Giovana Montoro Pazzini Watfe, Lais Fajersztajn, Euler Ribeiro, Paulo Rossi Menezes, Marcia Scazufca

**Affiliations:** 1Laboratorio de Investigacao Medica (LIM) 23, Instituto de Psiquiatria, Hospital das Clínicas HCFMUSP, Faculdade de Medicina, Universidade de São Paulo, São Paulo SP 05403-903, Brazil; giovana.pazzini@gmail.com (G.M.P.W.); scazufca@gmail.com (M.S.); 2Instituto de Estudos Avançados da Universidade de São Paulo, São Paulo SP 05508-060, Brazil; 3Universidade Aberta da Terceira Idade, Universidade do Amazonas, Manaus AM 69029-040, Brazil; unatieuler@gmail.com; 4Departamento de Medicina Preventiva, Faculdade de Medicina FMUSP, Universidade de São Paulo, São Paulo SP 01246-000, Brazil; pmenezes@usp.br

**Keywords:** aging, elderly, disability, comorbidity, primary health care, prevalence, quality of life

## Abstract

In Brazil and in most low- and middle-income countries (LMICs), information about how prepared the health care system is for the rapid aging of the population is scarce. We investigated the prevalence of disability and areas of life affected by disability among elders of the public primary health care in São Paulo and Manaus, Brazil. We investigated whether people with disability visited a primary care professional more frequently, the individual characteristics associated with disability, and differences by city. We randomly selected participants aged ≥60 years (*n* = 1375). The main outcome was disability, evaluated with the 12-item World Health Organization Disability Assessment Schedule (WHODAS 2.0). Exposure variables were consultation with a family physician, sociodemographic characteristics, health status, social support, and lifestyle. The prevalence of global disability was higher in Manaus (66.2% vs. 56.4% in São Paulo). In both cities, participation and mobility were the areas of life most affected by disability. The number of consultations with a family physician was not associated with disability. The high prevalence of disability and associated risk factors indicates that public primary health care is not meeting the needs of elders in both cities. It is warning because most elders in LMICs live in more underserved communities compared to Brazil.

## 1. Introduction

Disability is not a natural consequence of aging [1] and should largely be prevented and treated in primary health care settings [2,3]. Part of the prevention effort is effective screening for chronic diseases, which are more prevalent in older adults, and controlling other conditions that can lead to disability [2]. In Brazil, three-quarters of the population over 49 years of age has at least one chronic disease [4], and 23% of the individuals in that age group already have difficulty performing at least one activity of daily living, such as dressing themselves and getting up from a chair [5]. In addition to activities of daily living, disability affects several other areas important to independent life, such as work, leisure, and participation [6,7,8,9,10,11]. Disability increases the risk of mortality [12], the need for long-term care, and the risk of institutionalization, affecting patients and their families in economic, social, and emotional terms. The economic impact of disability on health care systems is significant [13]. This is a warning sign for the Brazilian unified health system (Sistema Único de Saúde (SUS)), given that Brazil has one of the most rapidly aging populations in the world [14].

Within the elder population, the prevalence of disability is higher among women, the oldest age brackets, individuals with a lower level of education, and residents of low- and middle-income countries (LMICs), as demonstrated by studies conducted in several countries in Europe, Asia, Africa, the Americas, and the Middle East [5,8,12,13,15,16,17,18,19,20,21,22,23,24,25,26,27]. Knowing the functional status of the elder population is important to identify the most vulnerable members and their health needs. As a result, health planners can better define priorities for the allocation of resources and devise health policies that prevent the onset and worsening of disability, within the scope of primary health care [11].

In 2017, there were over 30 million individuals ≥60 years of age in Brazil and approximately two-thirds sought treatment for their health problems via SUS [4]. According to the Brazilian National Policy for Older Adults, primary care is the main point of entry into the public health care system and should prevent and treat most health conditions that can result in disabilities [2], as well as identify and treat older adults with disabilities. However, the SUS is not prepared to meet the growing demands of the elders, such as identifying and preventing disability or planning specific intervention programs targeting the associated risk factors [9].

In this study, we investigated the prevalence and factors associated with disabilities and areas of life affected by disability among older adults enrolled at primary care clinics (PCCs) affiliated with the Family Health Strategy (FHS) in two large metropolitan cities in Brazil: São Paulo and Manaus. We expect to find a similar prevalence of disability in both cities, as the population investigated lives in social disadvantaged areas and has their health needs covered mostly by the Brazilian Unified Health System. In addition, we also studied whether older adults with more disabilities, and consequently more health needs, were seen more often by primary health care professionals. We anticipate that having more disabilities will be associated with an increased number of PCC consultations in São Paulo and Manaus. Hence, we hope that our findings will collaborate with the planning and management of more effective primary health care services, contribute to improve public policies for older adults, and promote healthy aging in Brazil and other LMICs [7].

## 2. Materials and Methods

### 2.1. Study Design

This was a cross-sectional study based on data from a larger investigation on the health of older adults registered at PCCs in the cities of São Paulo and Manaus, Brazil, and described elsewhere [28].

### 2.2. Study Settings and Participants

The study was conducted with older adults registered at FHS-affiliated PCCs in the Brazilian cities of São Paulo and Manaus. São Paulo is the capital of the southern state of São Paulo and is 3885 km from Manaus, which is the capital of the northern state of Amazonas. The two cities have marked socioeconomic, demographic, and cultural differences. São Paulo has 12 million inhabitants, of whom approximately 1.3 million (12%) are ≥60 years of age; is well connected to other large cities in the state; and is surrounded by important industrial and financial centers. In contrast, Manaus has 2 million inhabitants, approximately 6% of whom are ≥60 years of age; is the most populous and wealthiest city in northern Brazil; is surrounded by the Amazon rainforest and rivers; and is isolated from other municipalities in the Amazon region.

The FHS is the main model of primary health care in the SUS, a universal and unified public health system free of charge [29]. The organization of the FHS is the same throughout Brazil. In September 2018, there were 42,960 FHS teams nationwide, collectively serving approximately 150,360,000 individuals of all ages, approximately 62% of the Brazilian population. FHS teams’ coverage in Manaus is lower than that in São Paulo, 28% and 32%, respectively. A more traditional model of primary health care covers the areas of the country where the FHS has not been implemented yet. Through its teams, FHS organizes the care of the families enrolled in the program. Each FHS team is composed of one family physician, one nurse, two nurse assistants, and up to 12 community health agents. Each team provides comprehensive, longitudinal care (health promotion, disease prevention, and treatment) for 3000–4000 individuals (~350 of whom are aged ≥60 years). The number of households covered by each family team depends on the vulnerability of the population living in the catchment area.

We contacted the primary health care coordinator in the two cities to obtain a list of the FHS-affiliated PCCs in the respective study areas. We then asked the managers of those PCCs to obtain a list of the names, and respective addresses, of all adults ≥60 years of age registered with the selected FHS teams, the study’s target population. In São Paulo, each PCC had up to seven FHS teams, whereas in Manaus each PCC had only one family team. For each PCC in São Paulo, we randomly selected two to four FHS teams. Next, we composed the study sample by randomly selecting 20 older adults from each FHS based on the following criteria: 11 women—60–69 years old (*n* = 5), 70–79 years old (*n* = 4), or ≥80 years old (*n* = 2); and 9 men—60–69 years old (*n* = 4), 70–79 years old (*n* = 3), or ≥80 years old (*n* = 2). These criteria reflect the gender and age group profile of the Brazilian older adult population. We also created a participant reserve list based on the same criteria. Whenever possible, individuals who declined to participate were replaced by individuals of the same gender and age group from the reserve list. Selecting participants using these criteria avoids under or over sampling individuals from specific gender and age groups (or obtaining almost no data from some gender and age group strata), as the number of individuals selected in each FHS team was small. Then, we attached weights to each sampled individual, based on the frequency in each FHS team (we used the list of all older adults registered in each FHS team), of individuals from the same gender and age group as the ones sampled.

Participants were interviewed in their homes by trained research assistants. Data were collected between 2010 and 2011. The research assistants used tablet computers to record the responses electronically.

### 2.3. Measures

The main outcomes of interest were global disability and severe disability. We evaluated disability with the 12-item World Health Organization Disability Assessment Schedule (WHODAS) 2.0 [30], a multicultural generic instrument based on the International Classification of Functioning, Disability, and Health [31]. The WHODAS has been translated into several languages and is largely used to study disability among adults in the community [13,30,32,33]. The WHODAS is adapted and validated for use in Brazil [33,34]. The WHODAS 2.0 evaluates six domains (areas of life): participation (joining in community activities), mobility (moving and getting around), life activities (domestic responsibilities, leisure, work, and school), cognition (understanding and communicating), self-care (hygiene, dressing, eating, and staying alone), and getting along (interacting with other people)—each of which is assessed with two items (questions). We used the short version of WHODAS (12 items), where each question is graded on a five-point Likert scale ranging from a score of 0 (no difficulty) to a score of 4 (extreme difficulty), with the exception of the question related to depressive mood, for which a score of 0 indicates no depression and a score of 4 indicates severe depression. The total score ranges from 0 to 48 [30]. Based on the literature [16,35,36], we generated two dichotomous (yes/no) variables to indicate disability: “global disability” and “severe disability”.

“Global disability” is defined by the presence of any difficulty in at least one of the 12 items of the WHODAS (total score ≥1). To generate the “severe disability” variable, the WHODAS total score was converted to a disability score ranging from 0 to 100. For each city, the WHODAS disability score was dichotomized as “yes” or “no”, according to whether or not it was above the 90th percentile. Therefore, 10% of the participants in each city were classified as having “severe disability”. While “severe disability” is valuable to investigate the characteristics that are associated with disability among the most impaired individuals within a specific population, “global disability” provides a more comprehensive picture, since it investigates the characteristics associated with having disability, no matter the severity.

We investigated sociodemographic, health status, lifestyle, and social support characteristics, using the following exposure variables: gender; age; years of schooling (0–3 vs. ≥4); marital status (married/partnered vs. single/divorced/widowed); personal income, (based on the national minimum wage 0, >0 ≤1, >1 <2, or ≥2); self-reported skin color/race (White, Asian, Black, Brown, or indigenous); self-reported morbidities (0, 1, or ≥2); depression (yes or no); self-reported health status (very good, good, satisfactory, poor, or very poor); consultation with a family physician in the last three months (yes or no); smoking (yes or no); opinion regarding the support received from children and friends (pleased or displeased); church attendance in the last year (none, occasional, or >3 times per month). We assessed depression with the nine-item Patient Health Questionnaire [37]. The presence/number of morbidities was assessed with a self-report questionnaire that includes the 20 most common health problems among older adults [38], such as hypertension, diabetes, arthritis, rheumatism, hearing problems, vision problems, asthma, bronchitis, stroke, and heart disease.

### 2.4. Statistical Analysis

Before conducting the statistical analyses, we assigned weights to the 20 individuals sampled in each FHS team. All analyses considered the design weights (sampling weights). We used descriptive analyses to compare the characteristics of individuals in the two cities. Because all exposure variables were categorical, we used the Rao–Scott chi-square test, which is a design-adjusted version of the Pearson chi-square test. We estimated the prevalence of global disability, calculating the corresponding 95% confidence interval (CI) for each city. We also estimated the prevalence of global disability for each exposure variable and the prevalence of all six domains of functioning (participation, mobility, life activities, cognition, self-care, and getting along with people), by gender. To describe the association of each characteristic (exposure) with global disability and severe disability, we constructed Poisson regression models with robust error variance, adjusted for gender and age, estimating prevalence ratios (PRs) and calculating the corresponding 95% CIs. We also used Poisson regression models with robust error variance to account for the complex survey sampling and because the outcome variables were common, with a prevalence >10% [39]. Although the sample was obtained with pre-established criteria for gender and age, the proportions of women and younger elders (60–69 years of age) were slightly higher than originally planned in both cities. To avoid confounding by gender and age, we adjusted the estimated PRs for the characteristics frequently associated with disability and for the other exposure variables investigated for gender and age. Data were analyzed with the STATA program, version 14.1 (StataCorp LP, College Station, TX, USA).

### 2.5. Ethics

All study protocols were approved by the Research Ethics Committee of the Faculdade de Medicina FMUSP, Universidade de Sao Paulo (reference no. 075/15). All participants gave written informed consent.

## 3. Results

We included 1375 participants: 702 in São Paulo and 673 in Manaus. The population size, considering the design effect, was 12,604,598 (design df = 37) in São Paulo and 68,687,509 (design df = 31) in Manaus. Less than 5% of the eligible individuals refused to participate, and five participants included in the original sample (one in São Paulo and four in Manaus) were excluded from the study because they were not assessed for disability. Participants were registered with 38 FHS teams from 13 PCCs in São Paulo and with 34 FHS teams from 34 PCCs in Manaus.

Table 1 summarizes the characteristics of the participants in São Paulo and Manaus, and the prevalence of global disability, according to sociodemographic, health status, lifestyle, and social support characteristics. In both cities, the proportion of women was slightly higher than the 55% originally planned. The same happened for the age groups, given that the original plan was that 45% of the participants would be in the 60- to 69-year age bracket, 35% would be in the 70- to 79-year age bracket, and 20% would be in the ≥80-year age bracket. The proportions of elder participants in the 70- to 79-year and ≥80-year age bracket were higher in Manaus than in São Paulo. Socioeconomic status was lower in Manaus than in São Paulo (lower level of education and personal income). In Manaus, a higher proportion of the population with disability was Black, Brown, or indigenous, compared to São Paulo. The frequency of self-reported morbidities was higher in Manaus than in São Paulo. In both cities, less than 9% of the participants reported being displeased with the support received from their children and friends. Church attendance was more frequent in Manaus than in São Paulo. The frequencies of depression, self-reported health status, smoking, and consultation with a family physician in the last three months were similar in both cities.

The prevalence of global disability was significantly higher in Manaus than in São Paulo—66.2% (95% CI: 62.6–69.6) vs. 56.4% (95% CI: 51.4–61.3), significant (*p* = 0.001). In Manaus, the criteria for global disability were met by at least three quarters of the individuals ≥80 years of age, who had ≥2 morbidities, had depression, or did not attend church (Table 1). In both cities, the same was true for individuals who perceived their health status as satisfactory, poor, or very poor. Despite the fact that less than 9% of individuals reported being displeased with the support received from children and friends, the prevalence of global disability among those same individuals was high in both cities, 90% in São Paulo and 82% in Manaus.

In all WHODAS domains, the participants in Manaus reported more problems than did those in São Paulo (Figure 1). In both cities, impairment was reported most frequently in the participation and mobility domains. In Manaus, 50.2% of the women and 42.3% of the men presented impairment in the participation domain, compared with 43.8% and 27.4%, respectively, in São Paulo. Problems in cognition were reported by 35.9% of the individuals in Manaus, compared with only 16.9% in São Paulo. The ranking of the WHODAS domains by gender was similar in both cities, except for life activities and cognition among women, which ranked third and fourth, respectively, in São Paulo and fourth and third, respectively, in Manaus. In both cities, impairment in the participation, mobility, life activities, and cognition domains was more common among women than among men.

The presence of morbidities and a self-perceived health status of satisfactory, poor, or very poor were significantly associated with global and severe disability in both cities (Table 2). The risk of global disability was higher among the individuals with ≥2 morbidities compared to those without morbidities in São Paulo (PR = 1.50; 95% CI: 1.08–2.09) and in Manaus (PR = 1.74; 95% CI: 1.46–2.06). In São Paulo, the risk of severe disability was twice as high among the individuals with ≥2 morbidities than among those without morbidities (PR = 1.91; 95% CI: 1.09–3.35), whereas that risk was four times higher in Manaus (PR = 4.64; 95% CI: 2.16–9.93). Among individuals who had a negative perception of their health status, the risk of severe disability was considerably higher than among those who had a positive perception of their health status—in São Paulo (PR = 3.46; 95% CI: 2.18–5.50) and in Manaus (PR = 2.70; 95% CI: 1.63–4.45). In that same comparison, the risk of global disability was approximately half of that of severe disability—in São Paulo (PR = 1.78; 95% CI: 1.48–2.15) and in Manaus (PR = 1.41; 95% CI: 1.23–1.61). Gender was associated with disability only in São Paulo, where men were found to be less likely to present global disability than women. The risk of global disability and the risk of severe disability were higher among older adults in both cities, but severe disability was not significant in São Paulo. Depression was associated with severe disability. The risk of severe disability among individuals with depression was approximately twice higher in Manaus (PR = 3.52, 95% CI: 1.95–6.35) than in São Paulo (PR = 1.41; 95% CI: 1.26–1.58). Being displeased with the support received from children and friends was associated with a higher risk of global disability and severe disability in both cities, but the difference was not statistically significant in Manaus. The risk of severe disability was significantly higher for individuals who had not attended church in the last year prior to the study, compared to those who had attended church at least three times per month in the same period in both cities. Some characteristics of participants decreased the risk of global disability, such as being male in São Paulo (PR = 0.74; 95% CI: 0.63–0.87) and having ≥4 years of schooling in Manaus (PR = 0.88; 95% CI: 0.78–0.99). Individuals who were current smokers in São Paulo were more likely to be at risk of global disability than were those who were non-smokers (PR = 1.43; 95% CI: 1.10–1.85). Consultation with a family physician in the last three months was not associated with disability in either city.

## 4. Discussion

We evaluated disability among adults ≥60 years of age registered at PCCs in São Paulo and Manaus and found marked differences in the prevalence of disability and of the associated factors in both cities. The prevalence of global disability was generally high, although it was higher in Manaus than in São Paulo (66% vs. 56%). The WHODAS domains in which the individuals in both cities most often reported impairment were participation and mobility, such impairment being more prevalent in Manaus and among women in both cities. A self-perceived poorer health status and having ≥2 morbidities were associated with global disability and with severe disability in both cities. Other characteristics associated with disability varied by city or by the degree of disability (e.g., depression was associated only with severe disability in both cities). Consultation with a primary care physician in the last three months before the study was reported by 48% of the interviewees in São Paulo and by 42% of those in Manaus, although it was not significantly associated with disability in either city.

The prevalence of global disability found in both cities is in line with the results of other studies of elder populations using a definition of global disability similar to that employed in the present study. In those studies, conducted in the Russian Federation, Mexico, South Africa, Ghana, China, India, Cuba, the Dominican Republic, Venezuela, and Peru, the reported prevalence of global disability ranged from 43% to 93% [16,20,36]. A previous study, conducted in São Paulo [13], found the prevalence of global disability to be 62.2%, higher than that observed in the present study, although that study included individuals ≥65 years of age, rather than ≥60 years of age as in the present study. Similarly, in São Paulo and Manaus, the prevalence of global disability was higher than that reported in studies investigating younger populations [5,23,25,35], which is to be expected because disability is age dependent [12,20].

As in studies that compared the prevalence of disability between countries [12,16], the fact that the prevalence of global disability was higher in Manaus than in São Paulo might be attributable to differences in population characteristics. In comparison with older adults in São Paulo, those in Manaus were more socially disadvantaged (lower level of education and lower personal income) and presented more morbidities, characteristics that are well known to increase the risk of disability [17,20]. Although we investigated only two cities in Brazil, our results indicate that we can expect the prevalence of global disability to be higher among older adults living in communities with higher level of social disadvantage, in Brazil and worldwide.

Apart from socioeconomic and other differences between São Paulo and Manaus, we can speculate that differences in the prevalence of global disability reflect differences between the two cities in terms of the quality of and access to PCCs. In both cities, the older adults were served by PCCs with FHS teams that follow the same model of organization. However, the management of PCCs is decentralized, which can lead to divergences in the delivery of care [29]. In fact, the PCCs in São Paulo are usually better equipped than are those in Manaus, with pharmacies that stock a wider range of medication and examination rooms that are more prepared to treat people. In addition, it is likely that the access to care in Manaus is hampered by prejudice on the part of health professionals, as shown in a previous study, in which health professionals working at PCCs in Manaus were found to be more likely to stigmatize older adults with depression than were their counterparts in São Paulo [28].

In the present study, the WHODAS domains in which the individuals in both cities most often reported impairment were mobility and participation, as previously reported in studies conducted in Spain, Poland, Iran, and rural villages in India [8,17,24,27,40]. The WHODAS participation domain assesses engagement in community activities and presence of emotional problems, whereas the mobility domain assesses physical difficulty in getting around, such as difficulty standing for long periods and walking long distances [30]. The presence of problems in participation domain may represent a mobility problem, given that the score on this domain is dependent on the ability of elders to move about independently whenever they have no support from their families to engage in social activities outside of the home. The other question accessed in this domain is about emotional problems, including being sad or depressed, which may also be highly affected by mobility problems in older adults.

In our study, it was not possible to establish the direction of the association between depression (measured with the PHQ-9) and severe disability, but a recent Polish study that used the extended 36-item version of WHODAS 2.0 [30] to measure disability among 1800 older adults did [41]. In this study, depression was significantly associated with moderate and severe disability but not with mild disability [41]. In the same study, the adjustable risk of depression among older adults with moderate or severe disability was significant for all domains, and the domains more severely affected were participation and mobility [41].

However, prioritizing depression management in primary care, as suggested in other studies [16,28], might be a reasonable strategy to manage disability and improve participation, as one of the main symptoms of depression is lack of interest or pleasure in activities. The high prevalence of impairment in the WHODAS participation and mobility domains might also be related to the high prevalence of ≥2 morbidities among the individuals with global disability in the population studied (73% in São Paulo and 83% in Manaus). These associations draw attention to the need for better screening for morbidities among older adults at the primary care level, in order to prevent disability.

In accordance with evidence from a wide range of studies conducted in Europe, Asia, Africa, the Middle East, and Latin America, our study indicated that being female and being in the oldest age brackets increase the risk of disability [8,12,20,24,25,26,27]. Our data show that global disability and severe disability were both significantly associated with multiple morbidities and the self-perception of a poorer health status, as previously been shown by studies in other populations [7,12,16,17,18,26,42]. However, we found that consultation with a family physician in the last three months was not associated with global disability or with severe disability. In other words, a relevant proportion of the participants had seen a physician (approximately half), but participants with disability were not more likely to have seen a physician than those without disability. This result suggest that older adult disability is not a primary health care priority in any of the cities studied and should serve as a warning sign, because it might indicate that the health needs of disabled older adults are not being met. Information about morbidities, gender, and age, the characteristics associated with disability in this study is easily available in primary health care, and asking about self-perception of health status is simple and fast. Looking for these characteristics is more feasible than applying specific disability scales such as WHODAS 2.0. FHS team members could routinely screen for disabilities in older adults exhibiting these warning signs (being female, being ≥80 years of age, having ≥2 morbidities, and having a self-perception of poor health status). That could help prevent disabilities and the worsening of existing disabilities.

Our study has some limitations. First, due to the cross-sectional study design, it is difficult to make inferences from our data. However, in certain circumstances, we can discuss causation. For example, elder people with severe disability in both cities were less likely to have attended church in the last year before the study. Because it is unlikely that church attendance prevented disability, we can speculate that disability decreased church attendance and possibly other social activities. In addition, church attendance depends on mobility and is an activity that represents social engagement, which underscores our finding that mobility and participation were the two WHODAS domains in which impairment was most common. Another potential limitation is the fact that we assessed disability using the short (12-item) version of the WHODAS, rather than the 36-item version recommended by the WHO [30]. However, a previous study that used the short and long versions of the WHODAS presented results similar to ours [43]. There is also evidence that the 12-item version of the WHODAS explains 81% of the variance identified with the 36-item version, which might validate our results [30]. Overall, the analyses of association presented good accuracy, indicating that our results can be interpreted with confidence. Finally, to reduce the likelihood of selection bias, we randomly selected the Family Health Teams from of each PCC and the older adults registered with these PCCs. In addition, to increase the variability of other individual characteristics of participants that may have an effect on the estimate of the prevalence of disability (e.g., socioeconomic background), we included participants registered with a large number of FHTs in São Paulo (38 FHTs from 13 PCCs) and Manaus (34 FHTs, each PCC in Manaus has only one FHT). However, our sample frame present limitations as the findings we presented may not be completely generalized to a small group of the older adults who mostly use the private health care sector. It is important to note that approximately two-thirds of the older Brazilian adults use exclusively the public health system for treatment and that individuals that use the private sector can use the public system. Given the importance of the primary health care in Brazil (either the Family Health Strategy model or the previous PCC model), we believe that our findings are relevant to the majority of the older Brazilians.

## 5. Conclusions

The characteristics of the older adults associated with disability in both cities were poorer self-perceived health status and having ≥2 morbidities. This information is easily available in primary care and can help identifying people with disability-related needs.

The FHS is organized to offer comprehensive treatment to patients and their families and to promote quality of life. However, the results of the present study show that this model of primary health care is not prepared to deal with the rapid aging of the population of Brazil and, consequently, with the increased prevalence of disability. Gaining access to primary health care consultations does not seem to be a major problem, given that 48% and 42% of the study participants in São Paulo and in Manaus, respectively, were seen by a family physician in the last three months before the study. Contrary to what we had expected, we found no association between disability and a greater frequency of consultations with the family physician. The FHS was designed to provide preventive and curative care, which includes meeting the demands of the most fragile elders and providing home care, services that were probably not being delivered as necessary in the two cities studied. In addition, the elder individuals showing the greatest disability were also those who were the most displeased with the level of support received from their family and friends. The abandonment of older adults by families and by the health care system, likely leads to a worsening of quality of life over time. It is of note that the socioeconomic and health indicators found to be associated with greater disability among the mostly low-income elder individuals evaluated in the present study were the same in both cities (education, income, morbidities, and skin color/race). Those indicators should be taken into consideration by health planners when they are deciding how to allocate the health care resources available, in order to promote the health of the older adults in Brazil according to their needs.

## Figures and Tables

**Figure 1 life-10-00133-f001:**
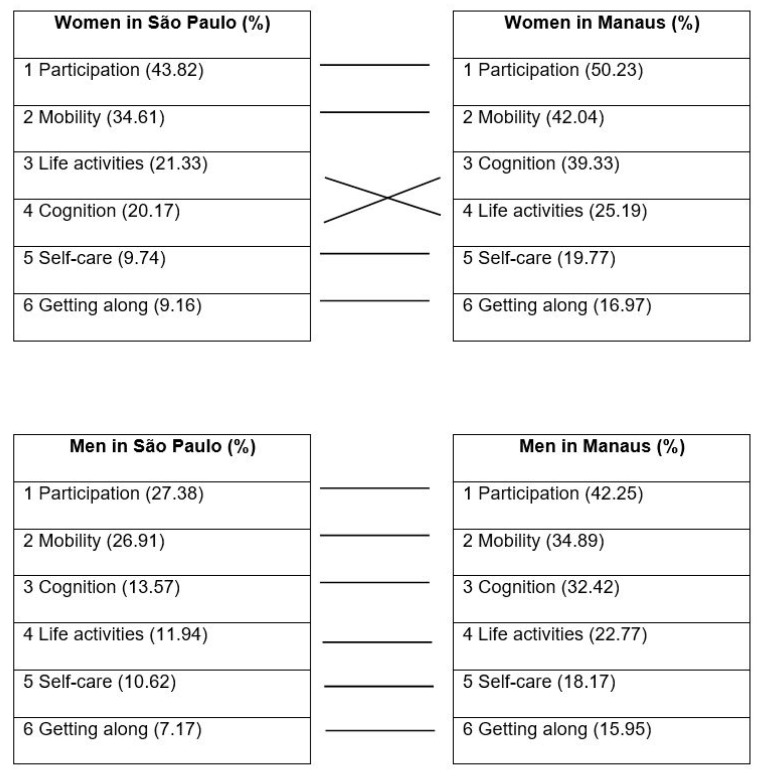
Prevalence of difficulty in the six domains of functioning of the World Health Organization Disability Assessment Schedule 2.0. Among older adults in the Brazilian cities of São Paulo and Manaus, by gender. Difficulty is defined by a score ≥1 in the domain.

**Table 1 life-10-00133-t001:** Characteristics of older adults with disability registered at primary care clinics in the Brazilian cities of São Paulo and Manaus.

	São Paulo		Manaus		
	(*n* = 702)		(*n* = 673)		
Characteristics	(%)	Global Disability (%)	(%)	Global Disability (%)	*p*-Value *
**Sociodemographic**					
**Gender**					
Female	61.11	62.6	57.99	69.3	0.01
Male	38.89	46.5	42.01	61.9	
**Age (years)**					
60–69	56.59	51.3	52.53	58.5	0.002
70–79	31.42	62.3	32.08	71.1	
≥80	11.99	65.0	15.39	82.9	
**Schooling (years)**					
0–3	50.71	56.8	61.92	70.7	0.018
≥4	49.29	55.0	38.08	59.4	
**Married/partnered**					
Yes	52.41	52.3	44.82	67.3	0.023
No	47.59	61.0	55.18	65.3	
**Personal income (*n* × MW) ^†^**					
0	19.51	56.7	14.36	53.4	<0.001
>0 ≤1	22.55	70.0	46.15	72.1	
>1 <2	26.62	54.9	24.0	63.9	
≥2	31.31	47.8	15.49	63.6	
**Skin color or race**					
White/Asian	51.60	60.7	29.91	60.6	<0.001
Black/Brown/Indigenous	48.40	51.2	70.09	68.4	
**Health Status/Lifestyle**					
**Morbidities (*n*)**					
0	47.12	48.1	34.56	46.4	<0.001
1	29.02	56.5	25.93	67.3	
≥2	23.87	72.7	39.51	82.9	
**Depression**					
No	92.22	55.9	92.21	63.9	0.996
Yes	7.78	62.7	7.79	91.3	
**Self-reported health status**					
Very good/good	56.12	41.4	49.84	54.7	0.103
Regular/bad/very bad	43.88	75.4	50.16	77.7	
**Smoker**					
No	87.77	53.2	90.86	67.1	0.141
Yes	12.23	69.8	9.14	66.4	
**Consultation with family physician** **(last 3 months)**					
No	51.9	53.0	57.86	62.6	0.177
Yes	48.1	60.0	42.14	70.8	
**Social Support**					
**Pleased with support received from children or friends**					
Yes	91.28	53.6	94.83	65.7	0.033
No	8.72	90.2	5.17	82.0	
**Church attendance in the last year**					
>3 times/month	42.71	54.0	62.33	64.1	<0.001
Occasional	41.87	57.8	28.91	66.3	
None	15.41	59.5	8.76	81.8	

* Comparison of the frequency of exposures between São Paulo and Manaus. ^†^ National minimum wage at the time of the study, in Brazilian currency Real (R$): R$510.00. Note: MW, Brazilian national minimum wage in 2010.

**Table 2 life-10-00133-t002:** Associations of global disability and severe disability with characteristics of older adults registered at primary care clinics in the Brazilian cities of São Paulo and Manaus.

Characteristics	Global Disability				Severe Disability			
	São Paulo		Manaus		São Paulo		Manaus	
	PR * (95% CI)	*p*-Value	PR * (95% CI)	*p*-Value	PR * (95% CI)	*p*-Value	PR * (95% CI)	*p*-Value
**Sociodemographic**								
**Gender (ref: female) ^†^**	0.74 (0.63–0.87)	0.001	0.89 (0.77–1.03)	0.135	0.66 (0.35–1.23)	0.190	0.84 (0.50–1.40)	0.512
**Age (years**)^†^	1.02 (1.00–1.02)	0.001	1.01 (1.00–1.02)	<0.001	1.02 (0.99–1.06)	0.068	1.03 (1.00–1.06)	0.048
**Schooling (ref: 0–3 years)**	1.03 (0.83–1.27)	0.748	0.88 (0.78–0.99)	0.043	0.95 (0.54–1.67)	0.858	1.53 (0.93–2.52)	0.086
**Married/partnered (ref: yes)**	0.97 (0.80–1.18)	0.83	0.87 (0.76–1.00)	0.060	0.48 (0.21–1.09)	0.081	0.80 (0.42–1.53)	0.500
**Personal income (*n* × MW)** **^††^**								
0	1	0.054	1	0.822	1	0.347		0.007
1 × the MW	1.12 (0.84–1.50)		1.19 (0.93–1.53)		1.09 (0.44–2.68)		1.35 (0.42–4.31)	
2 × the MW	0.92 (0.71–1.19)		1.10 (0.87–1.38)		0.95 (0.50–1.82)		0.83 (0.26–2.59)	
≥3 × the MW	0.87 (0.68–1.13)		1.13 (0.84–1.51)		0.68 (0.24–1.90)		0.48 (0.17–1.31)	
**Skin color/race (ref: White/Asian)**	0.28 (0.14–0.57)	0.001	1.13 (0.96–1.33)	0.131	0.73 (0.44–1.22)	0.232	0.75 (0.43–1.32)	0.311
**Health Status/Lifestyle**								
**Morbidities (*n*)**								
0	1		1		1		1	
1	1.17 (0.90–1.52)	0.217	1.38 (1.11–1.72)	0.004	0.98 (0.45–2.11)	0.955	2.32 (0.83–6.46)	0.104
≥2	1.50 (1.08–2.09)	0.017	1.74 (1.46–2.06)	<0.001	1.91 (1.09–3.35)	0.024	4.64 (2.16–9.93)	<0.001
**Depression (yes)**	1.05 (0.79–1.39)	0.701	0.76 (0.29–1.98)	0.564	1.41 (1.26–1.58)	<0.001	3.52 (1.95–6.35)	<0.001
**Self-perceived health status (ref: Regular/Bad/Very bad)**	1.78 (1.48–2.15)	<0.001	1.41 (1.23–1.61)	<0.001	3.46 (2.18–5.50)	<0.001	2.70 (1.63–4.45)	<0.001
**Smoker (yes)**	1.43 (1.10–1.85)	0.008	1.02 (0.80–1.32)	0.820	1.62 (0.66–3.98)	0.281	1.65 (0.85–3.22)	0.132
**Consultation with family physician in the last 3 months (ref: yes)**	1.12 (0.97–1.31)	0.113	1.10 (0.97–1.24)	0.107	1.01 (0.60–1.71)	0.948	0.97 (0.67–1.41)	0.887
**Social Support**								
**Pleased with support received from children or friends (ref: yes)**	1.64 (1.39–1.92)	<0.001	1.28 (1.03–1.59)	0.023	2.50 (1.47–4.26)	0.001	1.63 (0.57–4.65)	0.344
**Church attendance in the last year**								
≥3 times/month	1		1		1		1	
Occasionally	1.13 (0.94–1.35)	0.162	1.02 (0.93–1.12)	0.575	1.09 (0.61–1.95)	0.765	1.76 (0.88–3.51)	0.106
None	1.11 (0.83–1.50)	0.442	1.18 (1.00–1.38)	0.043	3.04 (1.73–5.33)	<0.001	4.41 (2.34–8.33)	<0.001

* Estimated by Poisson regression, with robust variance, and adjusted for gender and age, except where otherwise indicated. ^†^ Unadjusted prevalence ratio. Notes, PR, prevalence ratio; CI, confidence interval. ^††^ MW: Brazilian national minimum wage in 2010.

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
