# Peer review of "Prevalence of Older Adult Disability and Primary Health Care Responsiveness in Low-Income Communities"

_life, 2020, doi:10.3390/life10080133_

Round 1
Author Response
Thank you very much for your valuable suggestions. We will comment all of them bellow.
1) Title: The title is correct as it reflects correctly the objective and hypothesis of the work.
No comment.
2) Summary: This section follow a well structured format.
No comment.
3) Keywords; Please use recognised MeSH terms as this will assist others when they are searching for information on your research topic. The following website will provide these (simply start typing in a keyword and see if it exists or find an alternative if it does not): https://www.ncbi.nlm.nih.gov/mesh
We agree that Mesh terms as keyword is important and strategic. We selected new keywords and now all of them are recognized Mesh terms.
Keywords: aging, elderly; disability; comorbidity; primary health care; prevalence; quality of life (line 30)
3) Introduction: The research question itself is sound and the topic is strongly introduced. Please describe the hypothesis in this section.
We reviewed the last paragraph of the introduction and added what we expected to find regarding the main study questions and likely reasons for these results (lines 59-69). The current investigation is descriptive and thus no a proper a priori hypothesis has been formulated.
“In this study, we investigated the prevalence and factors associated with disabilities -and areas of life affected by disability- among older adults enrolled at primary care clinics (PCCs) affiliated with the Family Health Strategy (FHS) in two large metropolitan cities in Brazil: São Paulo and Manaus. We expect to find a similar prevalence of disability in both cities, as the population investigated lives in social disadvantaged areas and have their health needs covered mostly by the Brazilian Unified Health System. In addition, we also studied if older adults with more disabilities, and consequently more health needs, were seen more often by primary health care professionals. We anticipate that having more disabilities will be associated with increased number of PCC consultations in São Paulo and Manaus. Hence, we hope that our findings will collaborate to the planning and management of more effective primary health care services, contribute to improve public policies for older adults and promote healthy aging in Brazil and other LMIC [7].”
4) Materials and Methods: Please, describe the inclusion and exclusion criteria appear sound. Participants are reported to have consented appropriately and the protocol was approved by an ethics committee.
No comment.
5) Results: The results is clear and concise with appropriate statistical analysis been performed appropriately and rigorously.
No comment.
6) Discussion: The discussion appears well developed and appropriate, authors describe the results, the limitations and compare with other researchs. The limitations section needs to incorporate the sampling bias given the study design.
We incorporated in the discussion section (lines 347-358) limitations related to selection bias and the effect of the sample frame on our findings.
“Finally, to reduce the likelihood of selection bias, we randomly selected the Family Health Teams from of each PCC and the older adults registered with these PCCs. Besides it, to increase the variability of other individual characteristics of participants that may have an effect on the estimate of the prevalence of disability (e.g. socioeconomic background), we included participants registered with a large number of FHTs in São Paulo (38 FHT from 13 PCC) and Manaus (34 FHTs, each PCC in Manaus have only one FHT). However, our sample frame present limitations as the findings we presented may not be completely generalized to a small group of the older adults that uses mostly the private health care sector. It is important to note that approximately two-thirds of the older Brazilian adults use exclusively the public health system for treatment, and that individuals that use the private sector can use the public system. Given the importance of the primary health care in Brazil (either the Family Health Strategy model or the previous PCC model), we believe that our findings are relevant to the majority of the older Brazilians.“
7) Conclusion: The conclusion is conclusively.
No comment.
8) References: Appropriate
No comment.
9) Figures and tables: Correct.
No comment.
Reviewer 2 Report
This is a robust scientific study that is hypothesis driven. The results have impact for the design of health systems. The theoretical analysis could be deepened with regard to social science impliations, notably the relations between the domains of participation and mobility on the one hand and geographical characteristics on the other hand, and the impact of primary health physicians.
Author Response
Thank you for reviewing our manuscript and for your comments.
Reviewer 3 Report
It’s a very good database for more understanding of disability, and issues the author mentioned are interesting, but here are some issues author need to be clarified.
- The topic title of the article may need to reconsider, I can’t get the point.
- What are purposes of the study? The article only mentioned what they did, but what is the study for? For example, do the author want to compares the difference between two cities in disability? Or just for understand the incidence the disability rate of two cities? If the answer is “just for understand the incidence the disability rate of two cities: like sociodemographic, frequently of seeing primary care professionals……” the title of this article may need be modified. (my suggestion)
- What are the assumptions of the study (about low income, disability and primary care and their relationship)? In other words, what are the questions the study trying to answer?
- Detail question
- Line 75-81 is Redundant, you have mentioned in line 59-64
- 133 mentioned Global disability is defined by the presence of any difficulty in at least one of the 12 items of 133 the WHODAS (total score ≥ 1), I have some worry about this definition. As the manual of WHODAS 2.0 said, (page 44), you can see that in normal population, only 50% have no problem in WHODAS2.0. Which means even “normal” people, the WHODAS2.0 score would not be 0, this definition may lead Overestimate of disability rate. Can you do some explain for your decision?
- Have the data showed in Table 1 been weight? I can’t get it if your sampling is follow Stratified sampling by gender and age(line 108-110), why you need weight? Or I miss something?
- In Figure 1, I guess this “disability” means “any domain score ≥ 1”, am I right?
- Your data (Table 2) find some difference interesting factors that affect the disability rate between these two difference city, for example, the Smoker may relate to the disability in SÃO PAULO but not in MANAUS, do you have any ideas? You may discuss about these differences.
- Line 297 about the depression: The participation items in WHOPDAS2.0 12 version only have 2 questions, and only one of them is about engagement in community activities. The relation between patriation and depression may affect by the other item of WHODAS2.0 12 version (S5. How much have you been emotionally affected by your health problems?). you need expose that.
- Line 326: Disability may cause by many reasons, if we follow the ICF model. I don’t understand the primary health care function and purples in Brazil, they may fix other problems. I guess you need talk more about that.
Author Response
Thank you very much for your valuable suggestions. We have carefully reviewed your comments and have revised the manuscript accordingly.
1) The topic title of the article may need to reconsider, I can’t get the point.
The Title of the manuscript has been reviewed, as you suggested “Prevalence of older adult disability and primary health care responsiveness in low-income communities”. (lines 2-4)
2) and 3)
- What are the purposes of the study? The article only mentioned what they did, but what is the study for? For example, do the author want to compares the difference between two cities in disability? Or just for understand the incidence the disability rate of two cities? If the answer is “just for understand the incidence the disability rate of two cities: like sociodemographic, frequently of seeing primary care professionals……” ” the title of this article may need be modified. (my suggestion)
- What are the assumptions of the study (about low income, disability and primary care and their relationship)? In other words, what are the questions the study trying to answer?
We reviewed the last paragraph of the introduction and clarified the purpose of the study. We also included to this paragraph what we expected to find regarding the main study questions (prevalence of disability in both cities and the association between disability and attendance to primary care professional), and the reasons of expecting such findings (lines 59-69).
“In this study, we investigated the prevalence and factors associated with disabilities -and areas of life affected by disability- among older adults enrolled at primary care clinics (PCCs) affiliated with the Family Health Strategy (FHS) in two large metropolitan cities in Brazil: São Paulo and Manaus. We expect to find a similar prevalence of disability in both cities, as the population investigated lives in social disadvantaged areas and have their health needs covered mostly by the Brazilian Unified Health System. In addition, we also studied if older adults with more disabilities, and consequently more health needs, were seen more often by primary health care professionals. We anticipate that having more disabilities will be associated with increased number of PCC consultations in São Paulo and Manaus. Hence, we hope that our findings will collaborate to the planning and management of more effective primary health care services, contribute to improve public policies for older adults and promote healthy aging in Brazil and other LMIC [7].“
4) Detail question:
a. Line 75-81 is Redundant, you have mentioned in line 59-64
We fully agree with you suggestion. We have deleted lines 75-81.
b. 133 mentioned Global disability is defined by the presence of any difficulty in at least one of the 12 items of 133 the WHODAS (total score ≥ 1), I have some worry about this definition. As the manual of WHODAS 2.0 said, (page 44), you can see that in normal population, only 50% have no problem in WHODAS2.0. Which means even “normal” people, the WHODAS2.0 score would not be 0, this definition may lead Overestimate of disability rate. Can you do some explain for your decision?
We agree with your comment that the WHODAS 2.0 dichotomous scoring system (yes/no) may slightly overestimate the prevalence of disability, as the positive score (yes) merges individuals with mild to extreme level of difficulty in any of the disability items. Even so, we considered the WHODAS 2.0 the best option for our study. Our decision was because the WHODAS 2.0 is widely used, allowing comparisons of our results with studies conducted in other low and middle income countries, and the items are easy to understand and to apply to individuals with a low level of literacy. Besides that, there is no WHODAS 2.0 population norms to estimate the prevalence of disability in older adults using the summary score (0-100). Several large studies used the dichotomous (yes/no) scoring scale of the WHODAS 2.0 to estimate the prevalence of disability. The paper published by Korff and cols. (2008) reports the prevalence of disability in 80,737 residents of 17 countries participants of the Mental Health Survey. The study by Souza et al (2009) reports the prevalence estimates for 15,022 residents of seven countries. A more recent study published by Naidoo et al in 2017, reports the prevalence of disability based on a national sample of adults in South Africa (n=4,974).
c. Have the data showed in Table 1 been weight? I can’t get it if your sampling is follow Stratified sampling by gender and age(line 108-110), why you need weight? Or I miss something?
Yes, you are correct. Data shown in Table 1 is weighted. We reviewed the paragraph (lines 95-111) as follows:
“We contacted the primary health care coordinator in the two cities to obtain a list of the FHS-affiliated PCCs in the respective study areas. We then asked the managers of those PCCs to obtain a list of the names, and respective addresses, of all adults ≥ 60 years of age registered with the selected FHS teams, the study target population. In São Paulo, each PCC had up to seven FHS teams, whereas in Manaus each PCC had only one family team. For each PCC in São Paulo, we randomly selected two to four FHS teams. Next, we composed the study sample by randomly selecting 20 older adults from each FHS based on the following criteria: 11 women—60-69 years old (n = 5), 70-79 years old (n = 4), or ≥ 80 years old (n = 2); and 9 men—60-69 years old (n = 4), 70-79 years old (n = 3), or ≥ 80 years old (n = 2). These criteria reflect the gender and age group profile of the Brazilian older adult population. We also created a participant reserve list based on the same criteria. Whenever possible, individuals who declined to participate were replaced by individuals of the same gender and age group from the reserve list. Selecting participants using these criteria avoids under or over sampling individuals from specific gender and age groups (or obtaining almost no data from some gender and age group strata), as the number of individuals selected in each FHS team was small. Then, we attached weights to each sampled individual, based on the frequency in each FHS team (we used the list of all older adults registered in each FHS team), of individuals from the same gender and age group as the ones sampled.“
d. In Figure 1, I guess this “disability” means “any domain score ≥ 1”, am I right?
Figure1. You are correct. “Global disability” is defined by the presence of any difficulty in at least one of the 12 items of the WHODAS (total score ≥ 1). To clarify, we added this information at the figure legend (line 220).
e. Your data (Table 2) find some difference interesting factors that affect the disability rate between these two difference city, for example, the Smoker may relate to the disability in São Paulo but not in Manaus, do you have any ideas? You may discuss about these differences.
We did not discuss this issue as in both cities the number of participants are two small.
f. Line 297 about the depression: The participation items in WHOPDAS2.0 12 version only have 2 questions, and only one of them is about engagement in community activities. The relation between patriation and depression may affect by the other item of WHODAS 2.0 12 version (S5. How much have you been emotionally affected by your health problems?). you need expose that.“
We are not sure if we understood what the reviewer asked about depression and the participation items of the WHODAS. We argued that impairment in the participation domain is likely to be related to having depression (not to the S5 item of the WHODAS 2.0 12-items). We also presented likely reasons for such effect. We did not conduct any analysis of association using specific items of the WHODAS 2.0. We consider that for such analysis a much larger sample is needed.
g. Line 326: Disability may cause by many reasons, if we follow the ICF model. I don’t understand the primary health care function and purples in Brazil, they may fix other problems. I guess you need talk more about that.
We explain in the introduction that 2/3 of the older Brazilians use exclusively the Universal Health System (public system) and that primary care is the main entrance to this system (lines 52-58). The Brazilian primary health care is responsible for preventing and treat most health conditions that can result in disabilities. We added to this sentence another role of the primary health care: identification and treatment of older adults with disabilities (lines 55-56). On the conclusion section, we show the importance of the Brazilian primary care system on older adults’ life, and for improving their quality of life and disabilities (lines 360-381).
Round 2
Reviewer 3 Report
Thank you for your response
Everything looks great , only one thing need to be clarify.
Line 297 about the depression: I know you argued that impairment in the participation domain is likely to be related to having depression.
How you measure participation domain? Use the WHODAS2.0 12item right? (This is what I know). I mean, Questions in WHOPDAS2.0 12 version in participation domain only have 2, one is S4 “How much of a problem did you have joining in community activities (for example, festivities, religious or other activities) in the same way as anyone else can?” the other is “S5. How much have you been emotionally affected by your health problems?”
My question is
- how you measure “participation domain”
- If you use the WHOPDAS2.0 12 version, which item(s) you use to represent “participation”?
- If you use the definition made by WHODFES2.0, the S4 and S5 is used to respondent “participation domain” , then the question S5 is talk about emotion, which means this will cause the relationship between Participation and Depression in this study.
If I’m right all you need is to mention that.
If I miss understanding, you just need clarify that (my questions)
Author Response
Thank you for asking again these questions. Possibly we did not fully understand your questions in our previous replay, and for this reason we have not answered them correctly. We hope this new reply answer your questions.
We measured the participation based on the S4 and S5 questions of the WHODAS 2.0 12-items. We agree that the previous version of the manuscript was confusing regarding the association between the participation domain and depression. Below are the changes we made in this new version of the manuscript:
- We included the information that the participation domain ‘assesses engagement in community activities and presence of emotional problems’ (line 296).
- We found a typo on line, and we replaced the word ‘morbidities’ by ‘mobility’(line 298-299).
- We included at the end of this paragraph the phrase ‘The other question accessed in this domain is about emotional problems, including being sad or depressed, which can also be highly affected by mobility problems in older adults (lines 301-307).
- To avoid confusion between what we discussed in the previous paragraph about participation domain and depression, we started a new paragraph and made it clear that we are discussing the association between ‘severe disability’ and depression, as measured by the PHQ-9. Note that we included the word ‘severity’. We did not investigate the association between the participation domain and depression (lines 304-305).
- We included to the sentence starting with ‘However prioritizing…‘ the explanation ‘as one of the main symptoms of depression is lack of interest or pleasure in activities‘ (311-313).